# Accumulation Rule of Sugar Content in Corn Stalk

**DOI:** 10.3390/plants12061373

**Published:** 2023-03-20

**Authors:** Jianjian Chen, Yunlong Bian, Zhenxing Wu, Xiangnan Li, Tingzhen Wang, Guihua Lv

**Affiliations:** 1Institute of Maize and Featured Upland Crops, Zhejiang Academy of Agricultural Sciences, Dongyang 322100, China; chenjj@zaas.ac.cn (J.C.); wuzhenxing.1991@163.com (Z.W.); 15996276620@163.com (X.L.); wtz1398932310@163.com (T.W.); 2Jiangsu Key Laboratory of Crop Genetics and Physiology, Co-Innovation Center for Modern Production Technology of Grain Crops, Key Laboratory of Plant Functional Genomics of the Ministry of Education, Yangzhou University, Yangzhou 225009, China; ylbian@yzu.edu.cn

**Keywords:** corn stalk, sugar content, accumulation rule, plant water content, micro-Ribonucleic acids, Corn

## Abstract

The primary parts of corn stalks are the leaves and the stems, which comprise the cortex and the pith. Corn has long been cultivated as an grain crops, and now it is a primary global source of sugar, ethanol, and biomass-generated energy. Even though increasing the sugar content in the stalk is an important breeding goal, progress has been modest in many breeding researchers. Accumulation is the gradual rise in quantity when new additions are made. The challenging characteristics of such sugar content in corn stalks are below the protein, bio-economy, and mechanical injury. Hence, in this research, plant water-content-enabled micro-Ribonucleic acids (PWC-miRNAs) were designed to increase the sugar content in corn stalks following an accumulation rule. High-throughput sequencing of the transcriptome, short RNAs, and coding RNAs was performed here; leaf and stem degradation from two early-maturing Corn genotypes revealed new information on miRNA-associated gene regulation in corn during the sucrose accumulation process. For sugar content in corn stalk, PWC-miRNAs were used to establish the application of the accumulation rule for data-processing monitoring throughout. Through simulation, management, and monitoring, the condition is accurately predicted, providing a new scientific and technological means to improve the efficiency of the construction of sugar content in corn stalks. The experimental analysis of PWC-miRNAs outperforms sugar content in terms of performance, accuracy, prediction ratio, and evaluation. This study aims to provide a framework for increasing the sugar content of corn stalk.

## 1. Introduction

China is a large agricultural country with abundant fiber resources of all kinds of crops. However, corn straw, as agricultural waste, is mostly burned for fuel. Its smoke contains a large amount of TSP (total suspended particulate matter) and SO_2_, causing serious atmospheric pollution. Hydrogen energy is considered as the most important clean energy in the 21st century. Since China’s energy deployment is more focused on coal than oil and natural gas, this has led to a shortage of hydrogen. Hydrogen production from biomass energy is a new method of hydrogen production, which has received extensive attention. As an important agricultural waste, the thermal efficiency of converting corn straw into hydrogen can reach more than 30%. Biohydrogen generation yield and energy conversion efficiency may benefit greatly from the buffering effects of a solution with a high buffer capacity [1]. The most effective method of transforming maize stalk into complete reducing sugar was investigated, including using alkaline heat pretreatment followed by hydrolysis. This study confirmed that corn straw contains a lot of cellulose, hemicellulose and lignin. In lignocellulose, the hydrolysate of cellulose is only glucose, while hemicellulose is a multi-branched heteropolymer containing sugar residues including hexose, pentose and glycolic acids. Hemicellulose is easier to hydrolyze than cellulose, and the change in pentose content can well reflect the hydrolysis of hemicellulose [2]. It has been suggested that treating maize stalks with ammonium bicarbonate using ultrasonic energy might increase the amount of reducing sugar extracted from the stem. An ultrasonic factor, a temperature factor, and a liquid/solid mass ratio factor were tuned using a response surface methodology [3]. Hydrodynamic parameters were used to standardize and depict the impact of rheology on the fermentation process. Maximum yeast growth was seen in both the shake flask and fermenter [4]. Due to its lignin barrier, lignocellulose, the most plentiful and renewable carbon resource, remains unexplored. In biological pretreatment, the lignin is broken down by white rot fungus (WRF), allowing the cellulose and hemicellulose layers to be mined [5]. The effects of optimal pretreatment models on total sugar production from Corn leaf and sorghum stalk include breaking down the lignocellulose biomass into oligosaccharides. The pretreatment model analyzed the accumulation factors, such as the amount of time before treatment [6]. This research aimed to maximize the sugar content from corn stalks. Biomass components such as cellulose, hemicellulose, and lignin were examined, and structural changes in treated and untreated corn stalks [7]. For photo-fermentative hydrogen generation in natural circumstances, where temperatures fluctuate widely, strain screening for high-temperature tolerance is essential. Therefore, transposon mutagenesis was used to identify a strain that can survive in warmer environments [8].

Due to the wide range of environmental conditions for maize cultivation, nutrient management is crucial in maize cultivation. However, the current nutrient-management methods often fail to achieve the desired yield. While making effective use of available nutrients, nutrient management at specific sites becomes more important for achieving the expected results [9]. Most maize plants only have one stalk, which extends straight from the ground. The corn’s genetics and growing conditions influence the stalk’s final height. Its leaves appear as the stem develops [10]. The average protein and calorie content of corn stalks baled after harvest falls short of what is needed to winter a beef cow. Forage tests are inexpensive and may provide important information when feeding forage [11].

Calculating crop yields and implementing precise irrigation systems are only two examples of the numerous agricultural uses that rely heavily on water content monitoring. Prior research has estimated corn stalks either at a high spatial or a high temporal resolution [12]. This research looked at two novel and very energy-efficient concretes made using corn stalks as plant aggregate, using either magnesium phosphate cement or ordinary Portland cement as binders [13]. Small interfering RNAs (miRNAs) are a kind of non-coding RNA which perform various important functions in plants, including development, control of hormonal control, cell signaling, and cellular membrane systems in response to both biotic and abiotic distress [14,15].

The sugar content of corn stalk is of a complex quantitative character. As for the sugar content of the whole stem, the sugar content of the stem is different in different growth stages. In terms of the sugar content of each internode of the stem, the sugar content of different internodes in the same life for a long time and the sugar content of the same internode in different growth periods are also different. Based on the above discussion, the challenge of improving sugar content in corn stalks using the plant water-content-enabled micro-Ribonucleic acids (PWC-miRNAs) was designed, and the contribution has been listed here.

This project aims to provide a framework for increasing the sugar content of corn stalks.The PWC-miRNAs were implemented, improving the sugar content in corn stalk.The experimental result was validated with accumulation counterparts regarding performance, accuracy, prediction ratio, evaluation, and efficiency.

The remaining studies are arranged as follows. Section 2 includes a literature review of studies which evaluate the existing method. Section 3 recommends a plan for PWC-miRNAs and their implications. Section 4 provides experimental analysis, and Section 5 provides a conclusion and prospects.

## 2. Impacts and Implications of Sugar Content in Corn Stalk

Jiang et al. [16] were the first to generate Propane produced by a picture in a thin, long tube Photo BioReactor (PBR) operating in continuous mode; corn-stalk pith is a cheap organic resource. Hydrogen generation and photosynthesis were measured in a tubular PBR as a function of hydraulic retention time. Changes in refractoriness, acidity, sugar concentration, and soluble metabolite content of the soluble metabolites products (SMPs) along the peritubular region of the cell. Data analysis also showed statistically significant differences in optical cell density, sugar content, and SMPs throughout the tubular PBR.

Baptista et al. [17] suggested and improved a method for xylitol’s long-term, eco-friendly biotechnological synthesis. Utilizing an experimental approach, varied amounts of enzyme and substrate were used to maximize xylitol synthesis in a simultaneous scarification fermentation (SSF) process using Saccharomyces cerevisiae, which had been genetically modified. The total xylitol production by corn-cob valorization was improved by integrating these eco-friendly technologies and improving the suggested method.

Zhao et al. [18] explained an increase in hydrogen generation from cornstalk hydrolysate; this work introduces a unique technique based on the inclusion of biochar formed from the residue of Cornstalk remaining after pretreatment and hydrolysis (RCPH). Hydrogen production was significantly boosted in batch experiments when RCPH a concentration was added. The proposed results constitute an economically and ecologically desirable closed-loop process and may eventually provide complete cornstalk use with reduced energy input.

Wang et al. [19] explained that steam explosion (SE) was used as a pretreatment for maize stalks. For the first time, anti-tyrosine activity was measured in the hydrolysate found in the liquor of water explosion after cleaning corn stalks. Component profiles of hydrolysate and model substance analyses indicated that phenolic substances are the primary tyrosine inhibitors in the hydrolysate. Thus, this study aims to assess and enhance the anti-tyrosine activity of hydrolysate produced by steam-explosion processing of corn stalks.

Adekunle et al. [20] proposed that corn stalk (CS), a common agricultural byproduct, could be used in power generation to prevent it from becoming an environmental burden and boost the energy supply. In this investigation, CS was subjected to compressed hot water. The maize stalk preheated to Celsius was the best performing pretreated corn stalk (PCS). Ethanol yield was raised to 31.06 g/L after the slurry was filtered. Hemicellulose breakdown was measured at 73.8% under ideal circumstances. Increased surface area and induced porosity were seen in photos of the PCS. X-ray diffraction research showed that the PCS crystallinity index improved. The application of the processing of lignocellulose materials for the mass manufacture of fuel-grade bioethanol appears promising.

Ibrahim et al. [21] introduced using cornstarch as a binding matrix; biohybrid films were fabricated from cornstalk fiber in sugar palm fiber (SPF) using the solution casting technique. The prepared items were tested for their physical, structural, thermal, and tensile qualities. The density, water absorption, and soluble nature of the generated films decreased very little. Water-vapor permeability was reduced when the reinforcing fiber was loaded into the water barrier, demonstrating that the water-barrier’s properties exhibited increased resistance to vapor passage. No discernible shift in degradation temperatures was observed, suggesting that the hybrid composites’ thermal stability improved only a little.

Sari et al. [22] detailed the results of an experimental examination into the impact of corn husk fiber (CHF) content on the mechanical characteristics, water absorption behavior, and swell ability of CHF composites in aquatic settings. The findings suggest that composites have longer immersion times, and higher CHF contents are associated with reduced mechanical characteristics. The degradation is caused by the composite’s high water absorption rate, which decreases the bonding contact between CHF.

Zhang et al. [23] introduced the generation of bio-hydrogen (H2) from renewable biomass, which has been recognized as a viable strategy for the generation of future alternative fuel. This research looked at the feasibility of producing hydrogen from fermentation using a hydrolysate of corn stalks (CS) which had been processed with an alkaline-enzymatic hydrolysis process beforehand. The Energy Sustainability Index (ESI) value was measured during two-stage fermentation by comparing the output to the input. According to the findings, a two-stage fermentation process has great promise as a practical method for extracting hydrogen from lignocellulose biomass.

Suopajärvi et al. [24] explained deep eutectic solvent (DES) treatments; wheat bran, corn stalk, and canola stem was utilized and evaluated for decolorization and micro-oscillation in this study. Five of the DES treatments were acidic, and one was alkaline. One possible reason for this distinction is that the acidic DES degraded the peptide plus wax, weakening the nanofibers’ ability to adhere to one another and form a network. In addition, DES analysis revealed notable deviations from the chemical composition of the lignin fractions isolated following acidic and alkaline DES treatments.

Yang et al. [25] proposed a pretreatment in ionic liquids (ILs) which has seen extensive application in the biomass industry. However, it is currently unclear how IL residues influence biomass digestion and fermentation. Since leftover ILs might block subsequent hydrolysis and fermentation processes, it is common practice to flush the biomass thoroughly with water following pretreatment. This paper reveals a feasible method for making fermentable sugar and L-lactic acid, which has the potential for use in industry as part of a lignocellulose biorefinery.

Sun et al. [26] introduced lignocellulose biomass, used to make biofuels, as the best replacement for fossil fuels. This has led to the direct use of ionic liquids in several research efforts aimed at pretreating lignocellulose before fermentation. Here, maize cobs and stalks were pretreated with an aqueous solution of the strong proton acceptor tetra butyl phosphorus hydroxide (TBPH). Additionally, the sugar yield remained even after reusing the TBPH aqueous solution. These findings point the way to the fermentation of additional biomass to produce the necessary products.

From the above discussion, challenging characteristics such as the protein, bio-economy, and mechanical injury in corn stalks are taken into consideration as significant for using an accumulation rule, such as in [16,18,26]. Further, this research discusses the PWC-miRNAs, which help to predict performance, accuracy, prediction ratio, and evaluation.

## 3. PWC-miRNAs and Its Discussion

The percentages of the components of the fermenting liquid include bioethanol, methanol, and alcohol. The amount of material lost during electrodialysis and active carbon treatment was determined by measuring the absolute dry cloth in the corn stalk. Corn stalks have a high value because their hemicelluloses can be fermented into butanol, acetone, and ethanol; the long fibers can be used to make paper, and products such as polyesters and epoxy resins may be made with the help of lignin and short fibers. The key to using corn-stalk sugar as fuel-ethanol raw material is the sugar content (≥10%) and sugar yield of corn stalk after normal harvest. At present, there are few varieties of corn suitable for fuel-ethanol production. The main reason is the low sugar content of the stalk. The annual output of corn straw in China is about 224 million tons. Most of these straws are directly burned in the field. Only a small amount of straw is returned to the field or used as feed, resulting in huge energy loss. If the sugar content of corn stalk after harvest meets the requirements of fuel-ethanol production, there are certain advantages to extracting fuel ethanol from the sugar content of corn stalk (Figure 1).

Manual harvesting involves farmers physically removing ears of corn off their stalks and placing them in bags. The husk is removed at the farm or barn, and the cobs are thrown away or sold separately from the corn. The three main machines used for mechanical harvesting are the corn snapper, the corn picker-husker, and the corn combine harvester. This device removes the ear of corn off the stalk but does not hull it. The cobs of corn that have had their husks peeled away are what the corn picker-husker is after. The corn on the cob that is brought into the barn may be sold as is, or it can be put through the shelling process, at which point the corn kernels can be sold separately for a greater price. The last kind of harvesting equipment used for corn is the corn combine harvester, which is an integrated and multipurpose harvester that can snap, shell, store, and crush straw all at once. Due to their high price and size, these machines are best-suited for big farms. We suggest economic and environmental implications be included in the maize production and farm waste collection objective-based optimization.

It is important to educate farmers and other agricultural workers about the harm they are causing to the environment, which can include taking action to curb the practice of burning agricultural trash in the open. Believe that social responsibility concerns will have little impact on the overall gains from farming. If verified, this might be used as an incentive for farmers to switch to less harmful means of disposing of agricultural waste, complementing the sting of punitive actions. This research aims to find a way to reduce polluting gases released by landfills holding waste products by optimizing the flow of maize kernels and field waste from corn crops to sugar content.

From the above discussion on the sugar content level in corn stalks, accumulation rules such as [16,18,26] need to be improved in several ways. Therefore, this introduces an opportunity for PWC-miRNAs, which help to predict corn-stalk evaluation as influenced by the low amount of protein, bio-economy, and mechanical injury, as discussed: PWC-miRNAs’ activation function may be seen as the collection of transfer functions applied to produce the desired output based on input and feedback, effectively deciding the sugar content in corn stalk.

Cellulose, hemicellulose, and lignin are the primary building blocks of straw and other lignocellulose feedstock (Figure 2). Since cellulose and hemicellulose may be immediately turned into chemicals such as furan and amino compounds, in addition to paper, they are valuable chemical raw materials. Many other bio-based products may be made by hydrolyzing cellulose and hemicellulose into digestion carbohydrates such as fructose and galactose. Products are categorized in Figure 1 based on how corn stalks are used.

However, folks frequently employ a specific software or only one element in the handling of lignocellulose feedstock, such as the tar problem in corn-stalk gasification, the creation of weak acids all through combustion, the automatic barrier of fiber in the lipase of threads, the ineffectiveness of cellulose adsorbent, and so forth. One explanation is that corn stalk and other lignocellulose feedstock do not have a good fractionation process. However, the abovementioned issues arise because the conversion technique is unique and basic. The lack of the notion of multilevel use makes it impossible to use biological or chemical approaches to collect the whole maize stalk or use all three key components simultaneously.

To achieve a breakthrough in the core processes involved in corn-stalk conversion, there has to be a significant development of solid residues considered in the context of process development theory. Further, significant advances are required in the crucial technology development procedure, systems engineering, and optimization involved in the bioconversion of lignocellulose feedstock for fuels. When it comes to raw-material pretreatment, the paper industry’s reliance on polluting methods (such as dry de-pithing, which creates air pollution, and wet de-pithing, which generates wastewater) and on the tried-and-true method of acid hydrolysis, which is inefficient and expensive, is to blame for much of the damage done to the environment. Using conventional technologies and methods to produce alcohol from starch requires a large amount of cellulose. The conversion efficiency of alcohol is poor, requires a lot of money, and the direct production cost is high. A classification method for multiple products was designed and used in many layers to deal with the above problems, caused by the diversity of corn-straw composition, through hydrogen explosion and solid fermentation.
(1)$$\mathrm{Corn}\;\mathrm{stalk}=\frac{\mathrm{Improving}\;\mathrm{sugar}\;\mathrm{contain}}{\mathrm{cellulose}-\mathrm{rich}\;\mathrm{weight}}\times 100\%$$
(2)$$\mathrm{Digeslibility}=\frac{\mathrm{Increasing}\;\mathrm{sugar}\;\mathrm{content}}{\mathrm{Cellular}\;\mathrm{weight}}\times 0.9\times 100\%$$

Stalk refers to a corn plant’s main stem (or body). The corn ear weighs a lot; thus, the stalk has to be strong so that nutrients may circulate freely up and down the plant. The excess photosynthetic carbohydrates in the stalk and leaves of late-season maize give it a reddish-purplish hue. Even while the plant could produce sugar via photosynthesis, it has very few or no kernels to increase the sugar content. There is a fundamental mismatch between the plant’s sinks and its sources. Only one or two ears of corn will develop on most corn stalks. Sweet corn with a delayed maturity time will produce two ears. The second ear often develops later in life and is smaller than the first. PWC-miRNAs are used to solve the issues in corn stalks, increasing the sugar-content level in corn stalks.

## 4. Experimental Analysis

The PWC-miRNAs effectively predict and validate the corn-stalk evaluation compared with the accumulation-rule data method based on performance, accuracy, prediction ratio, and efficiency, which are discussed as follows.

Dataset description: 100 corn-stalk data were taken from [27] for this experimental analysis. Grain yield, phenol, and fiber content each hectare, plant dry in the stalks, percent of lignin and cellulose in the pseudo-stem, break the power of the stalk, and length were also measured. There was increased green-leaf area and stalk density when fungicides were used.

Stalk density inputs were taken for this performance analysis of corn stalks. Glucose, lignocellulose, and phenol may all be converted independently (Figure 3). This pretreatment approach has also been made possible, and is a method for distributing the polymeric elements of cellulosic feed. This strategy has been successful at home and abroad, making it the most crucial high-value lignocellulose usage concept. Processes for producing lignin, a renewable commercial resource, on a massive scale that is both environmentally friendly and highly efficient is possible; however, this is hindered by the economic and technological hurdles mentioned above. The proposed PWC-miRNAs showed the highest efficiency value, a 95% improvement compared to other existing methods.

The corn-stalk efficiency refers to the protein and energy level required by beef cattle who wintered on 93% of corn stalks after harvest. Stalk breaking strength and height inputs were taken for the efficiency analysis of corn stalk (Figure 4). A forage test is a worthwhile investment, especially given the low cost of feeding forages of all types. On average, a single corn stalk will yield two ears, depending on the variety. Although cultivars vary in terms of their emphasis and what they provide, the first ear is usually stronger and of higher quality than the second. Compared to other existing methods, PBR and RCPH, the proposed method PWC-miRNAs is more elevated in efficiency ratio prediction.

Sugar is a simple carbohydrate and a member of the family of compounds with a sweet taste with a common chemical structure (Figure 5). Stalk lignin and cellulose percentage are the inputs taken for the accuracy of sugar content based on PWC-miRNAs. Sugar is packaged in a wide variety of formats. Sucrose, lactose, and fructose are the three most common forms of sugar. While glucose is essential for cell function, too much of it may be harmful. Note the details on a drink’s nutrition label, including the sugar content in percent. Corn stalks, which are 91% similar to other graminaceous plants, are robust and measure between meters in centimeters in diameter. Approximately, dried maize stalks may be harvested from an acre of land. Leaves and stems make up the bulk of a corn stalk, with the latter including the cell walls of the cortex and pith. Compared to other existing methods, PBR and RCPH, the proposed method, PWC-miRNAs, is higher in accuracy ratio.

Corn stalks provide beneficial nutrients; the protein content of fully developed corn stalks is highest in the leaves, then the tassels and stalks, and, finally, the bracts, which contain the least crude protein (Figure 6). Grain yield, hectares produced, and the amount of lignin and cellulose in the stalks are the inputs taken for corn-stalk prediction based on PWC-miRNAs. The highest concentration of oil fiber is found in the bark at 89%, followed by a somewhat lower concentration in the pith and, finally, a much lower concentration in the leaves. The crassest fat is found in the leaves and the least in the tassel. The nutritional content of the plant components decreases from the leaves and tassels to the stalks and bracts. Compared to other existing methods, PBR-RCPH and TBPH, the proposed PWC-miRNAs are higher in prediction tests, as shown in Figure 5.

A corn-stalk’s skin is around millimeters thick, and the mass ratio and fiber content are about 80%. Based on PWC-miRNAs, the input taken for evaluation analysis for corn stalks was increased grain yield, stronger stalks, and more straw output (Figure 7). Regarding mechanical strength, the rind is the strongest portion of the corn stalk. Its fiber structure is similar to jute and wheat stalk, making it a good raw material for making panels and paper. Making class-A particleboard, which is a medium-density fiberboard, from the corn-stalk cortex is possible. The price of raw materials is likely to increase if just the rind were used instead of the rest of the corn stalk. Compared to other methods now in use, the suggested PWC-miRNAs performs better.

Therefore, future work discusses creating a sugar content in corn stalks with PWC-miRNAs’ assistance to validate the performance, accuracy, efficiency, and prediction results.

## 5. Conclusions

From these methods, BBR, RCPH, and TBPH from corn stalk are not predicted; it is effective using PWC-miRNAs method, the advantages are expected correctly, and the experimental analysis is compelling. This study creates PWC-miRNAs to meet the growing needs of an accumulation rule increasingly relied upon for sugar content in corn stalks. Predicting agricultural water needs and managing irrigation efficiently relies heavily on real-time field-scale monitoring of plant water status. Previous research has estimated crop water content by measuring the reflectance spectra of samples taken from the field and brought into the lab using an indoor spectrometer. This study provided the first convincing evidence that miRNAs have a role in Corn’s metabolism and the accumulation of sucrose-related genes. The study also reveals the multiple molecular mechanisms at play in Corn stems and how they work together to control the buildup of sucrose. In this work, researchers pinpoint the roles of numerous MiRNAs involved in glucose uptake, sugar transportation, and glucose retention, both well-known and novel, and the genes they regulate.

## Figures and Tables

**Figure 1 plants-12-01373-f001:**
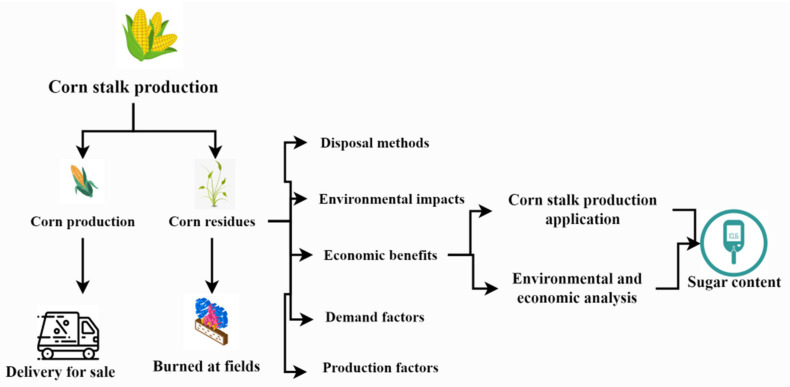
Sugar content in corn stalk.

**Figure 2 plants-12-01373-f002:**
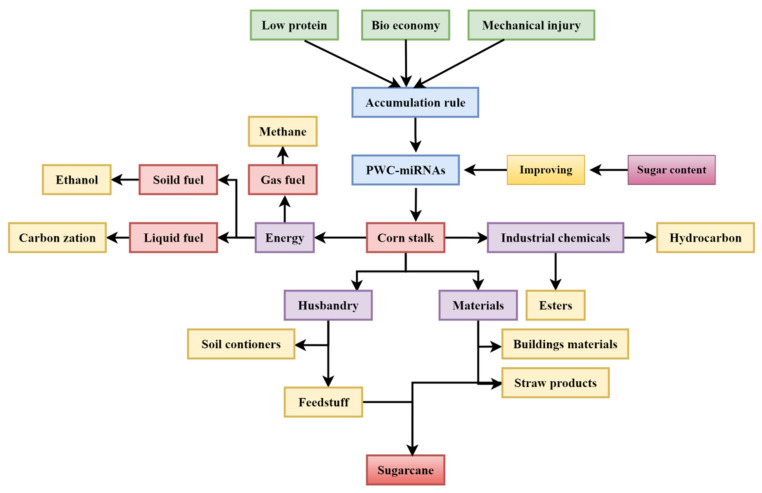
Conceptual framework implemented based on PWC-miRNAs.

**Figure 3 plants-12-01373-f003:**
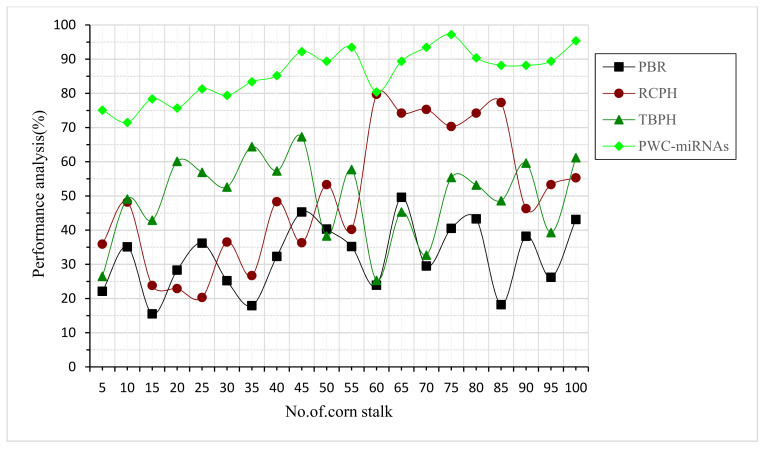
Sugar-content performance analysis of corn stalk.

**Figure 4 plants-12-01373-f004:**
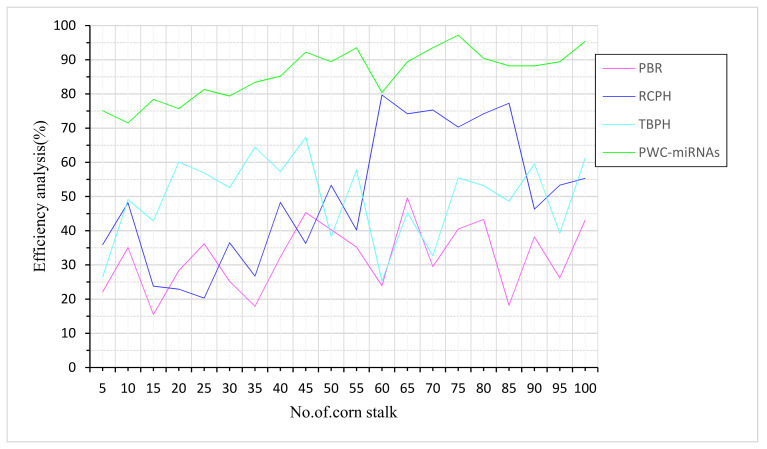
Efficiency analysis of corn stalk.

**Figure 5 plants-12-01373-f005:**
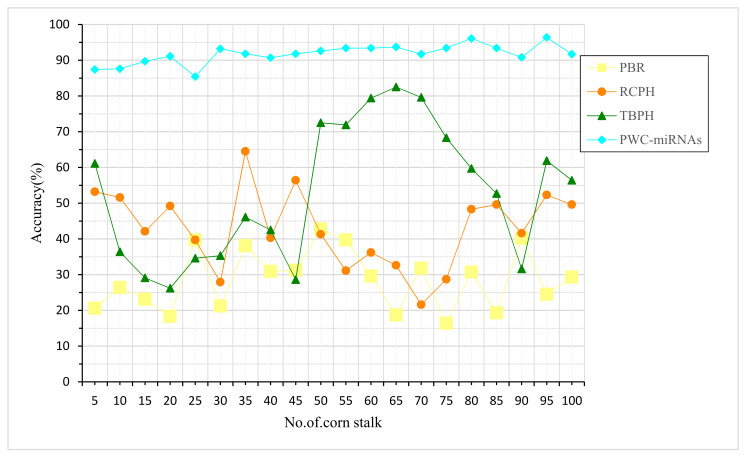
Accuracy of sugar content based on PWC-miRNAs.

**Figure 6 plants-12-01373-f006:**
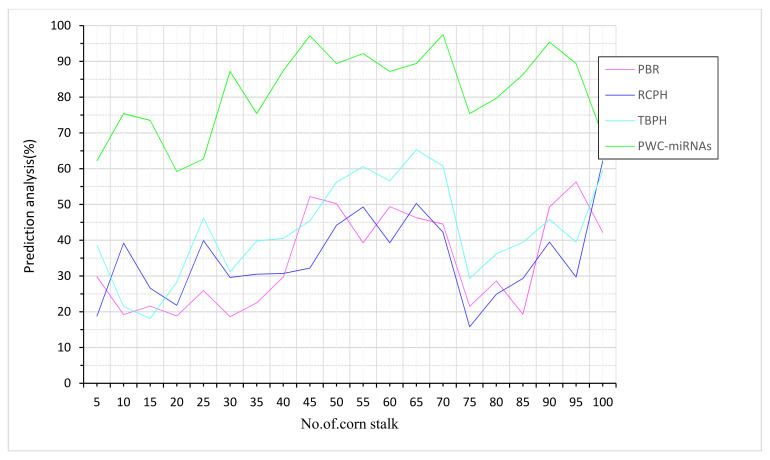
Corn-stalk prediction based on PWC-miRNAs.

**Figure 7 plants-12-01373-f007:**
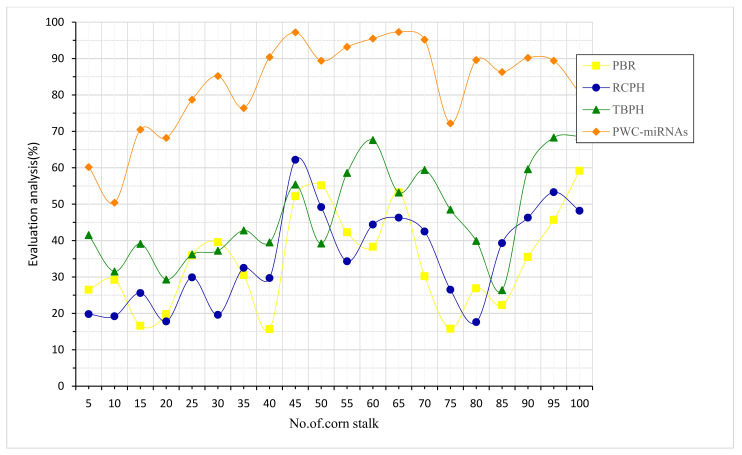
Evaluation analysis for corn stalk based on PWC-miRNAs.

## Data Availability

The figures used to support the findings of this study are included in the article.

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
