# Peer review of "Accumulation Rule of Sugar Content in Corn Stalk"

_plants, 2023, doi:10.3390/plants12061373_

Round 1

Reviewer 1 Report

The manuscript with the title "Accumulation rule of sugar content in corn stalk" is a well-written and organized work. It is properly structured and easy to follow. The authors provided sufficient information from specialized literature. These existing results are properly interpreted and can be published in their current form, with the recommendation to detail the purpose of this study in the abstract for better visibility.

Author Response

This study aims to provide a framework for increasing the sugar content of corn stalk.

Reviewer 2 Report

The experiment is well performed and the theme is, short, novel, and very interesting, but minor corrections are needed.

- Abstract, Line 12: change "Sugarcane" to "sugarcane".

- Introduction Line 13: change "Corn" to "corn".

- "Plant water content-enabled micro-Ribonucleic acids (PWC-miRNAs)" is repeated, it should only be used once and then PWC-miRNAs used throughout the manuscript.

- Change Jiang, D et al. to Jiang et al., and applied this modification on all citations in the text.

- For the title of Chapter 3, you can just write "PWC-miRNAs".

- Figure 1's resolution should be clearer.

- The background colors in some of the rectangles for Figure 2 should be lighter to increase the visibility of the words inside the rectangles.

- It is preferable not to start the sentence with the word “Figure” and to put it in brackets at the end of the sentence.

- For equations 1 and 2, and Figures 3-7, the font size must match the font size of the text.

- References are twice numbered and should be modified according to the journal's instructions.

- There are some grammar and typos in the text, please carefully read and revise.

Author Response

Corresponding modifications have been made according to the suggestions.
